# Ethylene Response Factor LlERF110 Mediates Heat Stress Response via Regulation of *LlHsfA3A* Expression and Interaction with LlHsfA2 in Lilies (*Lilium longiflorum*)

**DOI:** 10.3390/ijms232416135

**Published:** 2022-12-17

**Authors:** Yue Wang, Yunzhuan Zhou, Rui Wang, Fuxiang Xu, Shi Tong, Cunxu Song, Yanan Shao, Mingfang Yi, Junna He

**Affiliations:** Beijing Key Laboratory of Development and Quality Control of Ornamental Crops, College of Horticulture, China Agricultural University, Beijing 100193, China

**Keywords:** lilium, heat stress, LlERF110, LlHsfA3, LlHsfA2, ethylene

## Abstract

Heat stress seriously affects the quality of cut lily flowers. The ethylene response factors (ERFs) participate in heat stress response in many plants. In this study, heat treatment increased the production of ethylene in lily leaves, and exogenous ethylene treatment enhanced the heat resistance of lilies. LlERF110, an important transcription factor in the ethylene signaling pathway, was found in the high-temperature transcriptome. The coding region of LlERF110 (969 bp) encodes 322 amino acids and LlERF110 contains an AP2/ERF typical domain belonging to the ERF subfamily group X. *LlERF110* was induced by ethylene and was expressed constitutively in all tissues. LlERF110 is localized in the nucleus and has transactivation activity. Virus-induced gene silencing of *LlERF110* in lilies reduced the basal thermotolerance phenotypes and significantly decreased the expression of genes involved in the HSF-HSP pathway, such as *LlHsfA2*, *LlHsfA3A*, and *LlHsfA5*, which may activate other heat stress response genes; and *LlHsp17.6* and *LlHsp22*, which may protect proteins under heat stress. LlERF110 could directly bind to the promoter of *LlHsfA3A* and activate its expression according to the yeast one hybrid and dual-luciferase reporter assays. LlERF110 interacts with LlHsfA2 in the nucleus according to BiFC and the yeast two-hybrid assays. In conclusion, these results indicate that LlERF110 plays an important role in the basal thermotolerance of lilies via regulation of the HSF-HSP pathway, which could be the junction of the heat stress response pathway and the ethylene signaling pathway.

## 1. Introduction

Global climate warming limits the global productivity of crops [1]. Heat stress has negative impacts on the physiological and molecular functions of plant cells, such as cell membrane dysfunction, protein denaturation, and reactive oxygen species accumulation, resulting in oxidative stress and programmed cell death [2]. In response to heat stress, plants have evolved complex mechanisms to reduce heat stress damage, such as changes to their morphological characteristics, adjustments to their respiration and regulating metabolisms [3,4,5,6].

When faced with stress, the hormone levels in plants can change, which can regulate their metabolism and induce the formation of stress resistance [7]. Ethylene (ETH) is a gaseous plant hormone, which plays key roles in the vegetative and reproductive growth process of plants. Ethylene can accelerate respiration; regulate the differentiation of plant sexual organs; and promote fruit ripening, organ abscission, and senescence. In addition, ethylene also has effects on the abiotic stress response of plants [8,9,10,11]. The content of ethylene can increase when plants are subjected to abiotic stresses such as mechanical injury, waterlogging, drought, freezing, and heat stress [12,13]. Changes in the ethylene concentration can affect the reproductive growth of plants and induce the senescence of plant reproductive organs [14]. When facing heat stress, plants change the ethylene release in their reproductive organs to deal with heat stress and reduce damage [15,16,17,18]. In addition, the ethylene signaling pathway also interacts with other important thermal response signaling pathways to regulate plant heat tolerance [19,20,21].

The ethylene responsive factor (ERF) is an important subfamily of the AP2/ERF protein family, which contains an AP2 domain. ERFs are induced by ethylene and are the key transcription factors in the ethylene signal transduction pathway. They play important roles in abiotic stresses such as drought, flooding, low temperature, and salt stress [22]. In Arabidopsis, *ERF1*, *ERF74*, *ERF95*, and *ERF97* over expressing plants showed stronger heat tolerance than wild-type plants [23,24]. The *erf95 erf96 erf97 erf98* quadruple mutants showed a reduced basic thermotolerance; additionally, ERF95 and ERF97 interaction were shown to be induced by high temperatures [14]. The above studies showed that ERFs can be induced by ethylene in heat stress and play positive regulatory roles in plant thermal response.

Heat stress transcription factor (HSF) is a widely existing transcription factor in plants that regulates the expression of genes under high-temperature stress. HSFs are divided into three categories: HsfA, HsfB, and HsfC. HsfA plays significant roles in plants’ response to heat stress [25]. HsfA2 is a downstream response factor of HsfA1; overexpression of *LlHsfA2* was found to enhance the thermotolerance of transgenic Arabidopsis plants, and LlHsfA2 was found to interact with the LlHsfA1 protein to activate heat stress response genes after accumulation in the nucleus [26,27,28,29]. HsfA3 is also an important transcription factor in response to heat stress. Knockout or mutation of *HsfA3* can reduce the expression of the target *HSP* genes during heat stress [30]. Overexpression of *LlHsfA3A* in Arabidopsis can enhance both its basal and acquired thermotolerance [31]. Heat shock proteins (HSPs) are key stress proteins and are regulated by HSFs, which play an active function in the acquisition and maintenance of plant heat tolerance [32,33,34]. The HSF-HSP pathway is an important corresponding pathway of heat stress in plants, which plays critical roles in the production of plants’ heat tolerance [3,35,36].

Lily is globally known for its cut flowers, and it is native to temperate and cold regions in the northern hemisphere [37,38,39]. High temperatures in China reduce the quality and quantity of lily cut flowers [34,40]. Therefore, studying the molecular mechanism of heat stress response in heat-resistant lily varieties could provide important theoretical and practical significance for using genetic engineering technology to improve the heat resistance of the main existing lily varieties. In previous studies, we cloned several *HsfAs* in *Lilium longiflorum* ‘White Heaven’ such as *LlHsfA1*, *LlHsfA2*, *LlHsfA2b*, *LlHsfA3A*, *LlHsfA3B*, *LlHsfA4,* and found that they were all involved in heat stress response in lilies [27,28,29,31,40]. We also performed transcriptomic analysis regarding the heating of lilies for different lengths of time to search for other important genes, and we found the *LlERF110* gene [34]. In this study, we cloned the cDNA of ERF transcription factor *LlERF110* in *Lilium longiflorum* ‘White Heaven’, and found that it was induced by ethylene and heat stress. Virus-induced gene silence of *LlERF110* in lilies reduced the basal thermotolerance phenotypes and significantly decreased the expression of genes involved in the HSF-HSP pathway. LlERF110 could bind to the promoter of *LlHsfA3A* and interact with LlHsfA2. LlERF110 connects the ethylene signaling pathway and the HSF-HSP signaling pathway to jointly regulate the heat resistance of lily.

## 2. Results

### 2.1. Heat Stress Induced Ethylene Production

The endogenous ethylene production of lily leaves was tested after 3 h of treatments at 37 °C. The ethylene production of the lily leaves after heat stress was significantly higher than that of the control plants (Figure 1A). To study the effect of ethylene on plant heat tolerance, the phenotype of lily seedlings was examined by treatment with exogenous ethylene before heat stress. The lily seedlings were pretreated with 2 ppm ETH for 3 h and then exposed to heat stress at 37 °C for 26 h. The leaves of the pretreated lily seedlings were still green, while the leaves of the control plants had turned yellow (Figure 1B). These results showed that the pretreated lily seedlings were more heat resistant than those without pretreatment (Figure 1B). However, the lily seedlings did not show any phenotype under the pretreatment with 2 ppm 1-MCP (an inhibitor of ethylene receptors) for 3 h and then exposed to heat stress at 37 °C for 26 h (Appendix A). This may because other heat stress response pathways in lily weakened the negative effects of 1-MCP. These results indicated that heat stress could induce ethylene production in lily leaves, which may bind ethylene receptors and activate downstream signaling pathways to respond to heat stress. Therefore, the ETH treatment improved the heat tolerance of the lilies, but molecular regulation requires need further research with gene cloning and function analysis.

### 2.2. LlERF110 Was Induced by Heat Stress

An ethylene response factor gene, *LlERF110*, was found in the heat stress transcriptome of *Lilium longiflorum* ‘White Heaven’ tissue culture seedlings [34] (Appendix A). The transcriptome data showed that the expression level of *LlERF110* increased and reached the highest at 1 h and then decreased during the treatment at 37 °C for different lengths of time (Figure 2A). qRT-PCR was used to verify the results of *LlERF110* expression in the lily seedling leaves under the treatment at 37 °C at different times, and the results were consistent with the transcriptome data (Figure 2B). These results indicated that heat stress induces the expression of *LlERF110*, which may be related to the ethylene response pathway under high temperatures. Therefore, the characteristics and functions of *LlERF110* were explored in the subsequent section.

### 2.3. Molecular Cloning and Sequence Analysis of LlERF110

The *LlERF110* gene was cloned by designing primers according to the transcriptome sequence. The full-length DNA sequence of *LlERF110* contained two exons and one 381 bp intron; the open reading frame (ORF) of *LlERF110* was 969 bp, encoding a 322 amino acid protein (Figure 3A). The amino acid homology comparison showed that LlERF110 contained a conserved DNA-binding AP2/ERF domain, which is conserved in other plants such as *Daucus carota* var. *sativa*, *Arabidopsis thaliana*, *Helianthus annuus*, and *Arachis hypogaea* (Figure 3B). LlERF110 belonged to ERF group X (B-4) according to the amino acid homology comparison with 12 groups of ERF family genes of Arabidopsis [41] (Appendix A). The phylogenetic tree analysis showed that LlERF110 was closest to ERF110s in *Dendrobium catenatum*, *Cucumis melo*, *Musa acuminata* subsp. *Malaccensis*, and *Arabidopsis thaliana* (Figure 3C). These results indicated that the cDNA cloned from the lily was a homolog of the *ERF110* gene, whose function was analyzed under heat stress in lilium.

### 2.4. Expression Analysis of LlERF110

In order to study the role of *LlERF110* in responding to ethylene in lily, the expression of *LlERF110* was examined using qRT-PCR with treatment of exogenous ETH and ethylene receptor inhibitor 1-MCP. The expression of *LlERF110* increased significantly after the exogenous ETH treatment (Figure 4A) and decreased significantly after the 1-MCP treatment (Figure 4B). These results suggest that *LlERF110* is involved in the ethylene response signaling pathway in lily, whose expression was induced by the increase in ethylene concentration.

The temporal and spatial expression patterns of *LlERF110* were analyzed using qRT-PCR. The tissue expression of *LlERF110* was analyzed using different tissues of 120-day-old commercial lily plants (Figure 4C). *LlERF110* was expressed in all the tissues, and it was expressed the highest in the stem, followed by the stigma and the style; it was expressed the lowest in the petals. These results indicate that *LlERF110* could be constitutively expressed in all tissues and could play a role in the development of reproductive organs. Similarly, the expression of *LlERF110* was tested using the leaves of commercial bulb plants at different growth stages (Figure 4D). *LlERF110* was expressed in all the growth stages of the lily, and the expression was relatively low in the vegetative growth stage from 0 to 40 days, but significantly increased in the reproductive growth stage from 87 to 120 days. These results suggest that *LlERF110* could be expressed in all tissues and growth stages, which indicates involvement in the reproductive development stage of lily.

### 2.5. Subcellular Localization and Transactivation Activity of LlERF110

In order to verify the protein function, the subcellular localization of LlERF110 was explored in cells of tobacco leaves. A binary vector containing LlERF110::eGFP driven by the 35S promoter was constructed, then it was transformed into *Agrobacterium* and infected in *N. benthamiana* to observe the fluorescent signal. The green fluorescence of LlERF110::GFP was observed in the nucleus, which was overlapped with the red fluorescence of the nuclear localization marker NF-YA4-mCherry [42], while the GFP of empty vector 35S::GFP (control) was observed in the entire cell (Figure 5A). These results indicate that LlERF110 is localized in the nucleus.

LlERF110 may have a transcriptional activation function as ERFs are speculated to be plant transcription factors. The activation activity of LlERF110 was detected using a yeast system. BD, BD-LlERF110 and BD-GAL4 vectors were transformed into the yeast, respectively, and the cultured yeast was diluted to different concentrations, and growth was observed on the selected culture media. All the transformed yeasts of different concentrations could grow normally on the SD/-Trp plates. On the select SD/-Trp-His medium, the yeasts transformed with the BD-LlERF110 and the positive control BD-GAL4 of the recombinant plasmids grew normally at all dilutions. However, no growth occurred with the yeast transformed with the negative control BD plasmid. In the presence of X-α-Gal, the yeast transformed with the BD-LlERF110 and BD-GAL4 turned blue within 3 h, which means BD-LlERF110 activated the reporter gene *LacZ* of yeast allowing hydrolysis of X-α-Gal; yeast transformed with BD did not change (Figure 5B). These results revealed that LlERF110 has transcriptional activation activity.

The transcriptional activation activity of LlERF110 was examined using the LUC-REN system in *N. benthamiana*. The pattern diagram of the reporter and effectors were shown in Figure 5C. The effector plasmid of pBD-*LlERF110* was constructed; the empty vector pBD was used as the negative control and pBD-*VP16* was used as a positive control. The reporter plasmid containeds the yeast GAL4 DNA-binding element and the dual-luciferase (LUC) reporter, which was promoted by a minimal cauliflower mosaic virus *35S* promoter; we also used a *Renilla reniformis (REN)* reporter under the control of the 35S promoter as an internal control (Figure 5C). When co-infiltrated into *N. benthamiana* leaves with the reporter constructs, pBD-*LlERF110* resulted in higher firefly LUC activity than when pBD was used alone, although this activity was lower than with the use of positive pBD-VP16 (Figure 5D,E), suggesting that *LlERF110* is a transcriptional activator.

### 2.6. Virus-Induced Gene Silencing of LlERF110

In order to study the role of *LlERF110* in heat stress of lily, *LlERF110* was silenced in ‘White Heaven’ commercial bulbous plants, and the phenotype was observed at a high temperature. Using the silencing system, five *LlERF110*-silenced plants were obtained, and two strains (BSMV: *LlERF110-1* and BSMV: *LlERF110-13*) are shown in the image. The silenced plants were treated at 42 °C for 24 h and recovered for 16 days at room temperature. During the recovery time after heat stress, the leaves of the silenced lines showed blackening, burning, and wilting, while the leaves of control plants were a healthy green (Figure 6A). The expression of LlERF110 was decreased according to the qRT-PCR analysis (Figure 6C). The chlorophyll content of the silenced plants was detected in the leaves at 8 d of recovery and was lower than that of control plants (Figure 6B). These results showed that silencing of *LlERF110* could result in a heat-sensitive phenotype and serious damage to the plants, which indicates that *LlERF110* participates in heat stress response of lily.

The HSF-HSP pathway plays critical roles in plants’ heat stress response. In order to understand whether *LlERF110* is involved in the regulation pathway of HSF-HSP, the expression of *Hsfs* and *Hsps* in *LlERF110*-silenced plants was detected. The expression of *LlHsfA2*, *LlHsfA3A*, and *LlHsfA5* decreased significantly in the silenced lines, while the expression of *LlHsfA1* and *LlHsfA4* did not change (Figure 6D). The expression of *LlHsp17.6* and *LlHsp22* also decreased significantly in the silenced lines (Figure 6E). These results showed that expression of *LlHsfA2*, *LlHsfA3A*, *LlHsfA5*, *LlHsp17.6*, and *LlHsp22* were regulated by *LlERF110*, which indicated that they could be downstream of *LlERF110*. These results suggest that the silencing of *LlERF110* could lead to the downregulation of *Hsfs* and *Hsps* and that *LlERF110* is involved in the HSF-HSP pathway in lily.

### 2.7. LlERF110 Binds to the Promoter of LlHsfA3A

The expression of *LlHsfA3A* and *LlHsfA2* decreased most significantly in *LlERF110*-silenced plants, which means that *LlHsfA3A* and *LlHsfA2* may be the target of LlERF110. ERF95 and ERF97 directly bind to the promoter of *AtHsfA2* and regulate heat stress response genes in *Arabidopsis* [43]. Therefore, the yeast one hybrid system was used to test whether LlERF110 would bind to the promoters of *LlHsfA2* and *LlHsfA3A*. The TATA box in the *LlHsfA3A* promoter was truncated as *spLlHsfA3A* because the *LlHsfA3A* promoter has self-activation in yeast (Appendix A). All the transformed yeast cells grew normally on the SD/-Ura-Trp plates. On the SD/-Ura-Trp X-Gal plates, the yeasts transformed with pJG-LlERF110 and pLacZi-*spLlHsfA3A* turned blue, while the yeasts transformed with the negative control and the yeasts transformed with pJG-LlERF110 and pLacZi-*pLlHsfA2* did not (Figure 7A). The yeast experiment results showed that LlERF110 could bind to the promoter of *LlHsfA3A*. Similarly, the dual-luciferase reporter assay also showed that LlERF110 could activate the expression of *LlHsfA3A* in the *N. benthamiana* leaves (Figure 7C,D). We also tested the expression of *LlHsfA3A* in lily leaves treated with 2 ppm exogenous ETH for 3 h. The expression of *LlHsfA3A* increased significantly with the ETH treatment (Figure 7B) and was similar to the expression level of *LlERF110* (Figure 4A). These results indicated that LlERF110 could bind to the promoter of *LlHsfA3A* and promote *LlHsfA3A* expression.

### 2.8. LlERF110 Interacts with LlHsfA2

LlERF110 regulates *LlHsfA2* at the transcriptional level, although it cannot bind to the *LlHsfA2* promoter. Perhaps LlERF110 and LlHsfA2 interact at the protein level; this can be verified using bimolecular fluorescence complementation (BiFC) and a yeast two-hybrid (Y2H) assay. The green fluorescence was found in the nucleus and was overlapped with the red fluorescence of the nuclear localization marker NF-YA4-mCherry. However, the GFP was not observed in the control cells (Figure 8A), which means that LlERF110 interacted with LlHsfA2 in the cell nucleus. Similarly, the yeast two-hybrid assay also revealed the interaction between LlERF110 and LlHsfA2 (Figure 8B). LlHsfA2 is a transcription factor with self-activating activity (Appendix A), so we truncated its acting domain and called it sLlHsfA2. All the transformed yeast grew normally on the SD/-Trp-Leumedia. On the selected SD/-Trp-Leu-Ade-Hismedium, the yeasts transformed with AD-LlERF110 and BD-sLlHsfA2 of the recombinant plasmids were able to grow normally with different concentrations of dilution, while the yeasts transformed with the negative control AD + BD and AD + BD-sLlHsfA2 plasmid did not. In the presence of X-α-Gal, the yeast transformed with AD-LlERF110 and BD-sLlHsfA2 turned blue within 3 h, while the negative control did not (Figure 8B). These results indicated that LlERF110 interacts with LlHsfA2 in the nucleus.

## 3. Discussion

### 3.1. LlERF110 Is a Transcription Factor Involved in Thermotolerance in Lilies

Many molecular response mechanisms to heat stress have evolved in plants. In recent years, there have been an increasing numbers of studies on the response of ERF family proteins to heat stress in *Arabidopsis thaliana*, *Lycopersicon esculentum*, and other species [43,44,45,46,47,48,49], but not in lilies. In our study, *LlERF110* was selected with its expression changes in the transcriptomic data regarding the response of lily to heat stress (Figure 2). *LlERF110* was cloned and was found to encode a new ethylene response factor (Figure 3). LlERF110 has an AP2/ERF conservative domain (Figure 3B), belonging to group X (B-4) of the ERF subfamily (Appendix A). LlERF110 was closest to ERF110s in the monocotyledon according to the phylogenetic tree analysis (Figure 3C). The expression of *LlERF110* was induced by ethylene (Figure 4A) and reduced by 1-MCP (Figure 4B). The subcellular localization experiments and transactivation assay showed that LlERF110 was mainly located in the nucleus and had transcriptional activation activity (Figure 5). These results indicate that LlERF110 is a transcriptional activator and an ethylene response factor that could have functions in heat stress response in lilies.

The function of LlERF110 in heat stress response was verified using virus-induced gene silencing. The leaves of the LlERF110-silenced lines were wilted and withered, and the chlorophyll content was decreased upon heat stress at 42 °C (Figure 6A,B). The expression of LlERF110 was reduced in the silenced lilies, and the expression of several HSFs and HSPs genes was also decreased (Figure 6C–E), which is consistent with the heat sensitive phenotype. The tetraploid mutants *erf95 erf96 erf97 erf98* showed a decreased phenotype in basal thermotolerance in Arabidopsis [43]. Similarly, the *ERF1*-silenced plants showed a phenotype of high sensitivity to heat stress in tomato [49]. These results indicate that heat stress and ethylene induced the expression of LlERF110, which regulates the expression of HSFs and HSPs to participate in heat stress response in lilies.

### 3.2. LlERF110 Is Involved in Ethylene Signaling Pathway

Our studies showed that heat stress induced ethylene production in lilies, and the xogenous ethylene treatment enhanced the heat resistance of the lilies (Figure 1). 1-aminocyclopropane-1-carboxylic acid (ACC, a precursor to ethylene) added to plants protected them from heat-induced oxidative damage in Arabidopsis [50]. In a heat treatment experiment on pea seedlings, the ethylene production of the pea stems after heat treatment was higher than that of the control group [43]. After heat treatment, pretreated Arabidopsis seedlings with ACC had increased activities of antioxidant enzymes such as ascorbate peroxidase (APX), peroxidase (POX), superoxide dismutase (SOD), and catalase (CAT), as well as reduced oxidative damage, enhanced heat resistance, and improved survival rates as compared with the control group [45,47]. High-temperature stress induced the activities of ACC oxidase (ACO) and ACC synthase (ACS) and increased the endogenous ethylene release rate. These results show that ethylene could weaken the temporary damage caused by heat stress, which may enhance the antioxidant capacity in lilies.

In a study on Arabidopsis, in the presence of ethylene, ethylene binds to ethylene receptors and indirectly activate ETHYLENE-INSENSITIVE 3(EIN3)/EIN3-LIKE 1(EIL1), which can directly regulate downstream genes such as *ERF* and other genes to participate in the growth and development of plants [51,52,53,54,55,56]. The expression of tomato’s ethylene receptor downstream *LeCTRI*, rice’s key factor of ethylene signaling pathway *OsEIN2*, and Arabidopsis’ transcription factor *AtERF4* were induced by exogenous ethylene [57,58,59]. In our study, the expression of *LlERF110* was significantly increased after exogenous ethylene treatment, but it decreased after treatment with ethylene receptor inhibitor 1-MCP (Figure 4A,B), indicating that *LlERF110* is involved in the ethylene signal transduction pathway in lilies. In line with the above results, *LlERF110* may participate in the thermal response of lilies through the ethylene signal transduction pathway. Heat stress leads the accumulation of endogenous ethylene production in lilies, activates the ethylene signaling pathway, enhances the expression of *LlERF110*, and then participates in the heat stress response of lilies. However, the specific response mechanism of ethylene in heat stress response still requires further study.

### 3.3. LlERF110 Regulates the Expression of LlHsfA3A and Interacts with LlHsfA2

Heat shock transcription factors and heat shock proteins are important proteins that regulate high-temperature stress pathways in plants. In the virus induced gene silencing assay, the expression of *LlHsfA2*, *LlHsfA3A*, *LlHsfA5*, *LlHsp17.6*, and *LlHsp22* was significantly reduced in *LlERF110*-silenced lines (Figure 6), which means that the expression of these genes may be regulated by LlERF110. LlERF110 can bind to the promoter of *LlHsfA3A* and activate its transcription (Figure 7), indicating that LlERF110 is a direct upstream regulator of *LlHsfA3A*. The expression of *LlHsfA3A* was significantly increased with exogenous ethylene treatment (Figure 7B), which indicates that *LlERF110* regulates the expression of *LlHsfA3A* to participate in the HSF-HSP pathway in the heat tolerance response of lilies. Studies have shown that ethylene signaling mediates the HSF-HSP transcriptional pathway and regulates the expression of *HSFs* and *HSPs* genes under heat stress [23,24,47,49,60,61]. In a study on tomato plants, under the conditions of a high environmental CO_2_ concentration and high temperatures, the transcription levels of *HsfA2*, *Hsp70*, and *Hsp90* in *ERF1*-silenced plants were induced to increase, but they were still significantly lower than those in non-silenced plants [62]. This result is similar to that of previous reporter stating that ERF1 significantly induced the activation of the promoters of *HsfA3*, *Hsp101*, *Hsp70*, and *Hsp23.6* under high temperature conditions in *Arabidopsis thaliana* [23]. However, LlERF110 did not bind to the promoter of *LlHsfA2* (Figure 6B), which was not consistent with the result from the study on Arabidopsis. ERF95 and ERF97 directly bind to the *HsfA2a* promoter, and then regulate common heat response genes, such as *HsfA7a*, *Hsp90*, *Hsp20-like* and *Hsp17.6a*, to enhance the heat tolerance of Arabidopsis [43]. Perhaps the regulation between ERFs and HsfAs differs among different plant species. These results indicate that LlERF110 could regulate the expression of *LlHsfA3A* by directly binding its promoter in lilies.

The interaction between LlERF110 and LlHsfA2 was analyzed in this study. LlERF110 was found to interact with LlHsfA2 at the protein levels according to BiFC and the yeast two-hybrid assay (Figure 8). This interaction is the first reported for the first time, as the interaction between ERF and HsfA2 proteins has not been discussed in any previous studies. As LlERF110 increased the expression of *LlHsfA2, LlHsfA3A*, *LlHsfA5*, *LlHsp17.6*, and *LlHsp22*, the interaction between LlERF110 and LlHsfA2 probably enhanced the activity of LlHsfA2, which could regulate the expression of heat stress response (HSR) genes. This mechanism needs to be further analyzed.

According to the above results, *LlERF110* directly participates in the HSF-HSP regulatory pathway in heat stress response and is an important factor linking the ethylene regulatory pathway and the HSF-HSP pathway. Heat stress induced the accumulation of ethylene, which then induced the expression of *LlERF110*, directly binding the promoter of *LlHsfA3A* to regulate its expression and interact with LlHsfA2 allowing it to mediate the expression of heat stress response genes through stabilizing plant proteins, assisting in their renaturation, and preventing protein denaturation and aggregation (Figure 9). Further mechanisms by which *LlERF110* links these two pathways and regulates lily’s heat stress response needs to be further explored.

## 4. Materials and Methods

### 4.1. Plant Materials and Growth Conditions

Longiflorum hybrid ‘White heaven’ (*Lilium longiflorum*) plantlets were obtained by tissue culture. They were cultured on Murashige and Skoog (MS) basal medium at 22 °C in a standard culture room with a 16 h/light 8 h dark photoperiod. Commercial bulbous plants and seedlings were cultured in a nutrient medium containing peat soil and vermiculite medium at 25 °C/20 °C with 16 h light/8 h dark photoperiod. *Nicotiana benthamiana* was used as a model plant and was grown in a 16 h light/8 h dark photoperiod at 22 °C.

### 4.2. Detection of Ethylene Production, Exogenous Ethylene and 1-MCP Treatment

Lily leaves were treated at 37 °C for 3 h, then the leaves were cut off, weighed and placed in a 10 mL closed bottle, then incubated at 25 °C for 6 h to avoid the contamination of wound-induced ethylene. An amount of 2 mL of gas in the bottle was extracted using a gas-tight hypodermic syringe and injected into a gas chromatograph (SHIMADZU GAS CHROMATOGRAPH GC-17A), which was equipped with a flame ionization detector and an activated alumina column. and then we obtained the ethylene peak in the bottle [63,64]. According to the standard ethylene sample test, the ethylene peak occurred at about 0.5 min (Appendix A), so the peak appearing about 0.5 min in the sample was the ethylene peak and recorded its gas measuring area. The ethylene production was calculated according to the following formula:Ethylene production (nl ·g^−1^·h^−1^) = 
 Total sample volume (V1) × Gas measuring area (S) 
(58175.5 × Dry weight (DW) × Bottle sealing time (T) × Air inflow (V2)

Lily seedlings were pretreated in a closed glass box with 2 ppm ethylene and 1-MCP, respectively, for 3 h, and then treated for 26 h in an incubator at 37 °C for observation of the phenotype. Ten lily leaves were used for independent measurements and the average values are presented.

### 4.3. Gene Cloning and Sequence Analysis

Total RNA was extracted from ‘White heaven’ leaves using the RNAprep Pure Plant Kit (Vazyme, Nanjing, China) according to the manufacturer’s instructions. Then, 1 µg of RNA was subject to a reverse transcription reaction using the HiScript Q RT SuperMix Kit (Vazyme, Nanjing, China). The full-length sequence of *LlERF110* was cloned by designing primers (Appendix A) using transcriptome sequencing sequence. The conserved domain prediction was carried out using DNAMAN (Version 7) software and the NCBI website (https://www.ncbi.nlm.nih.gov/, accessed on 9 December 2021). The phylogenetic tree was analyzed using the TBtools v.1.098753 software and drawn using the iTOL website (https://itol.embl.de/, accessed on 23 June 2022).

### 4.4. Gene Expression Assay

In order to analyze the expression of *LlERF110* under high-temperature stress, the lily tissue culture seedlings were treated at 37 °C for 0 h, 0.5 h, 1 h, 3 h, 6 h, and 12 h. In order to analyze the effect of *LlERF110* on exogenous ethylene and 1-MCP, lily seedlings were treated for 3 h in a closed glass box containing 2 ppm ethylene and 1-MCP, respectively. After these treatments, RNA was extracted from leaves and the expression of *LlERF110* was detected by real-time fluorescent quantitative PCR. The leaves, bulb basal discs, roots, stems, styles, stigmas, petals, filaments and anthers of 120-day-old commercial lily bulbous plants were taken as materials. The RNA of different tissues was extracted and real-time quantitative PCR was used to detect the expression of *LlERF110*. RNA was extracted from commercial bulbous plants leaves of lilies at different growth stages, and *LlERF110* expression was detected using real-time fluorescent quantitative PCR. 18S rRNA was used as a quantification control, which has been validated in previous studies [40]. qRT-PCR analysis was performed using the 2^−∆∆CT^ method, and the primers designed for the qPCR analysis are listed in Appendix A.

For virus induced gene silencing assay, to analyze the expression levels of heat-related genes in wild-type and silenced plants, the leaves of commercial bulbous seedlings were collected at 10 days after injection, and the RNA was extracted for real-time quantitative PCR analysis. Each experiment included three biological replicates. All the relevant primers are listed in Appendix A.

### 4.5. Subcellular Localization of LlERF110

The ORF of LlERF110 was amplified by primers with *Xma*I and *Spe*I sites and cloned into a 35S-pCAMBIA2300-GFP-C vector to construct p35S::LlERF110-GFP, which was used for subcellular localization. For the tobacco injections, the binary vector was transformed into *Agrobacterium* GV3101 strains. The different agrobacterium lines were cultured in luria bertani medium, harvested by centrifugation at 3000× *g* for 10 min, and resuspended in the infiltration buffer (10 mM 2-(N-Morpholino)-ethanesulfonic acid, 10 mM MgCl_2_, 0.2 μM acetosyringone, pH 5.6) to a final optical density at 600 nm of ~1.0. Different suspension was mixed with nuclear localization marker NF-YA4-mcherry suspension and silencing suppressor P19 suspension, placed in the dark for 2–5 h and then infiltrated into *Nicotiana benthamiana* plants with 4–5 young leave [42]. The injected tobacco seedlings were then put in a chamber with a 16 h light/8 h dark light photoperiod at 22 °C. GFP images were obtained using a Zeiss LSM710 META confocal microscope with an excitation at 488 nm and an emission at 561 nm. Images were obtained after 48–72 h [65]. The primer sequences used for subcellular localization are listed in Appendix A.

### 4.6. Transcriptional Activity Analyses

The pGBKT7 vector (Clontech, San Jose, CA, USA) was used for the transcriptional activity analysis. The full length sequences of LlERF110 were amplified using primers with *Nde*I and *Eco*RI sites and then cloned into pGBKT7 to construct the pBD-LlERF110 vector. The vectors of pBD-LlERF110, pBD-GAL4 (positive control), and pGBKT7 (negative control) were transformed into AH109 yeast cells. They were screened on the SD/-Trp medium and SD/-Trp-His medium (used to detected whether the target protein activated the yeast reporter gene *HIS3*) and β-Gal activity (generated by the yeast reporter gene *LacZ*) was tested by X-α-Gal. The primers used in these experiments are listed in Appendix A.

### 4.7. Yeast One-Hybrid (Y1H) Assay

The 1021 bp and 356 bp promoter sequences of the *LlHsfA3A* promoter were amplified individually using PCR using primers with *Eco*RI and *Xma*I sites and then cloned into pLacZi to generate pLacZi-*pLlHsfA3A* and pLacZi-*spLlHsfA3A* (short promoter *LlHsfA3A*). The 723 bp sequence of the *LlHsfA2* promoter was amplified by PCR using primers with *Kpn*I and *Sal*I sites to generate pLacZi-*pLlHsfA2*. *LlERF110* was amplified using primers with *Eco*RI and *Xho*I sites and then cloned into pB42AD (pJG4-5) to generate pJG-LlERF110. The corresponding vectors were cotransformed into yeast strain Y187 to investigate binding. Successful transformants were selected by growth on SD media without Trp and Ura at 30 °C for 3 d. X-Gal is used to detect β-Galactosidase activity. Primer sequences used for Y1H are listed in Appendix A.

### 4.8. Dual-LUC Reporter Assay

For the transcription activity analysis, the coding sequence of *LlERF110* was amplified individually using PCR and primers with *Age*I and *Stu*I sites and then cloned into the pBD vector to construct pBD-*LlERF110* [66,67]. pBD-*VP16* was used as a positive control, and the empty vector pBD was used as a negative control [68].

For the transactivation assay regarding the binding of LlERF110 to the *LlHsfA3A* promoter, the 1021 bp promoter sequence of *LlHsfA3A* was amplified by PCR using primers with *Kpn*I and *Xma*I sites and then cloned into the pGreenII 0800-LUC vector (a plant expression vector containing *LUC* gene for detecting the binding between protein and promoter, usually clone the promoter sequence into this vector) to generate the *ProLlHsfA3A*:LUC reporter plasmid. The coding sequence of *LlERF110* was amplified by PCR using primers with *Xba*I and *Xma*I sites and then cloned into pGreenII 62-SK vector to construct Pro*35S:LlERF110* effector plasmids. The constructs were transformed into the *A. tumefaciens* strain GV3101 harboring the pSoup plasmid, respectively. The transformed *A. tumefaciens* lines were cultured in luria bertani medium with selection antibiotics. The cultivated agrobacteria were harvested by centrifugation at 3000× *g* for 10 min, and resuspended in the infiltration buffer (10 mM 2-(N-Morpholino)-ethanesulfonic acid, 10 mM MgCl_2_, 0.2 μM acetosyringone, pH 5.6) to a final optical density at 600 nm of ∼1.0. We mixed different combinations and placed them in the dark for 2–5 h, and then these were infiltrated into *Nicotiana benthamiana* plants with 4–5 young leaves. On the third day after infiltration, the ratios of LUC to REN were measured using the dual-LUC assay reagents (Promega, Madison, WI, USA) on a GLO-MAX 20/20 luminometer (Promega) [69,70]. Images of the LUC signals were captured using a CDD camera (CHEMIPROHT 1300B/LND, 16 bits; Roper Scientific, Sarasota, FL, USA). The primer sequences used for DLR are listed in Appendix A.

### 4.9. Yeast Two-Hybrid (Y2H) Assay

The coding sequence of *LlERF110* was amplified by PCR using primers with *Nde*I and *Eco*RI sites and then cloned into pGADT7(AD) to generate AD-*LlERF110*. *LlHsfA2* without the transcriptional activation domain was amplified using primers with *Nde*I and *Eco*RI sites and then cloned into pGBKT7 (BD) to generate BD-*sLlHsfA2*. The corresponding vectors were cotransformed into yeast strain AH109 to investigate the interactions. Successful transformants were selected according to growth on SD media without Trp or Leu at 30 °C for 3 d. We selected the strains that had successfully transformed and cultured them on an SD/-Trp-Leu-Ade-His medium to investigate their interaction. β-Gal activity was tested by X-α-Gal. The primer sequences used for Y2H are listed in Appendix A.

### 4.10. BiFC Assay

The coding sequence of *LlERF110* was amplified by PCR using primers with *Xba*I and *Xma*I sites and then cloned into pSPYCE (YCE) to generate YCE-*LlERF110*. The coding sequence of *LlHsfA2* was amplified by PCR using primers with *Spe* I and *Kpn* I sites and then cloned into pSPYNE (YNE) to generate YNE-*LlHsfA2*. All these vectors, along with the corresponding empty vectors as negative controls, were introduced into *Agrobacterium tumefaciens* strain GV3101, then cultured in Luria-Bertani medium with selection antibiotics. The cultivated agrobacterium was harvested by centrifugation at 3000× *g* for 10 min, and then re-suspended in the infiltration buffer (10 mM 2-(N-Morpholino)-ethanesulfonic acid, 10 mM MgCl_2_, 0.2 μM acetosyringone, pH 5.6) to OD_600_ = 1.0. We mixed different combinations and placed them in the dark for 2–5 h. The co-infiltration of different combinations of YNE and YCE with the nuclear localization marker NF-YA4-mcherry and the silencing suppressor P19 was carried out in *Nicotiana benthamiana* plants with 4–5 young leaves. After 72 h, the fluorescence signal was observed. The primer sequences used for BiFC are listed in Appendix A.

### 4.11. Verification of the Function of LlERF110 in Lily by the BSMV Silencing System

The barley stripe mosaic virus (BSMV) vectors (pCaBS-α, pCaBS-β, and pCaBS-γ) were used for virus-induced gene silencing [34,71]. The target fragment with a length of 278 bp of the *LlERF110* gene was cloned into the pCaBS-γ LIC (ligation independent cloning) vector and transformed into Agrobacterium spp. strain EHA105. Agrobacterium cells containing pCaBS-α, pCaBS-β, pCaBS-γ, or *LlERF110*-pCaBS-γ vectors were cultured in liquid Luria Bertani medium with selection antibiotics. The cultivated agrobacterium was harvested by centrifugation at 3000× *g* for 10 min, and then resuspended in the infiltration buffer (10 mM 2-(N-Morpholino) ethanesulfonic acid, 10 mM MgCl_2_, 0.2 μM acetosyringone, pH 5.6) to OD_600_ = 1.0. Then, suspensions of pCaBS-α, pCaBS-β, pCaBS-γ*-LlERF110*, or pCaBS-γ were mixed according to the ratio of 1:1:1 and placed in a dark environment at 28 °C for 3–5 h. Subsequently, the mixed Agrobacterium suspension was injected into the lily leaves using syringes without needles. Leaves of the lily plants were collected for qRT-PCR to verify gene silencing 8 days after injection. *LlERF110*-silenced plants were treated at 42 °C for 24 h and then recovered at 22–23 °C. The phenotype was observed, and photos were taken at different recovery times. The primers used are listed in Appendix A.

## 5. Conclusions

Heat stress can promote the production of endogenous ethylene in lily leaves that induces the expression of LlERF110, which mediates heat stress response via regulation of *LlHsfA3A* expression and interaction with LlHsfA2, and also plays an important role in the HSF-HSP pathway in lilies.

## Figures and Tables

**Figure 1 ijms-23-16135-f001:**
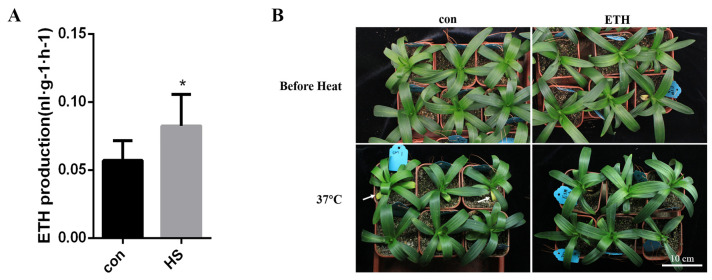
The production of ethylene in lily leaves under heat stress. (**A**) Examination of ethylene production of lily leaves after 3 h of heat stress at 37 °C. (**B**) Phenotype of lily under a heat stress of 37 °C for 26 h after pretreatment with 2 ppm ETH for 3 h. The white arrow points to the leaves that turned yellow. Bar: 10 cm *T*-test analysis of variance was employed to identify treatment means that differed statistically. Samples with stars are significantly different; * *p* < 0.05.

**Figure 2 ijms-23-16135-f002:**
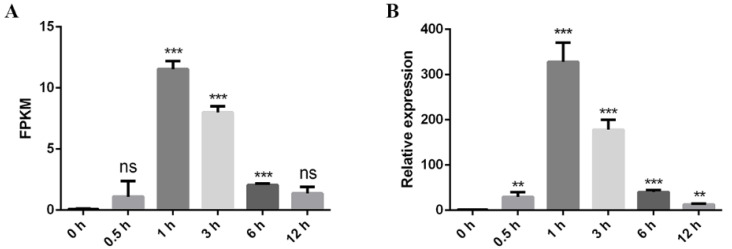
Expression of *LlERF110* under heat stress at different times. (**A**) RNA-seq data showing the expression of *LlERF110* under the 37 °C treatment at 0 h, 0.5 h, 1 h, 3 h, 6 h, and 12 h. (**B**) The relative expression of *LlERF110* was measured using qRT-PCR under the 37 °C treatment at 0 h, 0.5 h, 1 h, 3 h, 6 h, and 12 h. 18S rRNA was used as a control. Three independent experiments were performed each with three technical replicates; the results of one experiment are shown. *T*-test analysis of variance was employed to identify treatment means that differed statistically. Samples with different stars are significantly different; ** *p* < 0.01 and *** *p* < 0.001; ns means non-significance.

**Figure 3 ijms-23-16135-f003:**
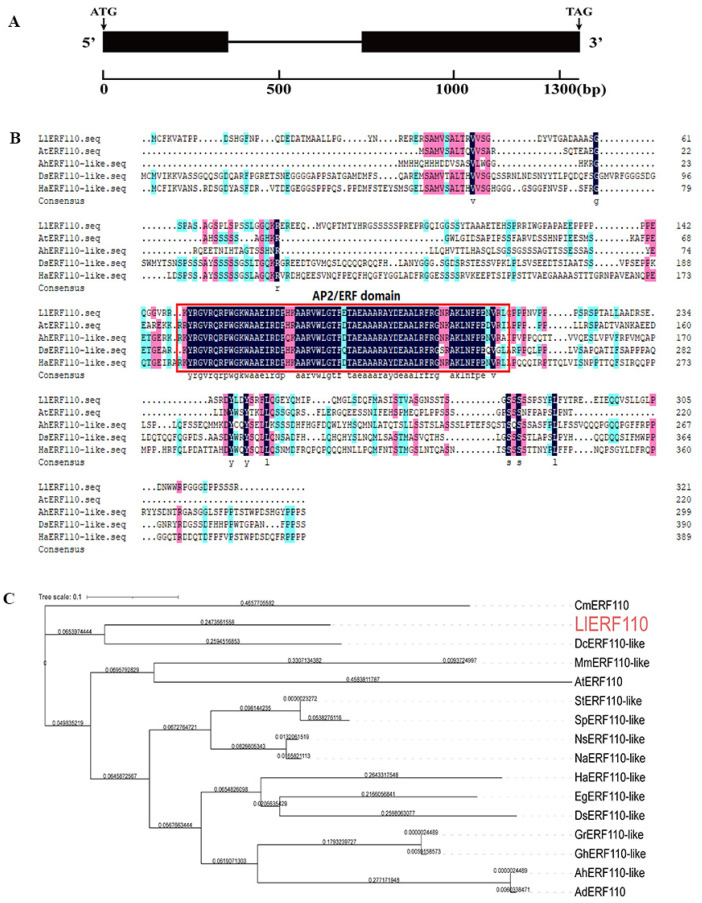
DNA structure and amino acid sequence of *LlERF110*. (**A**) DNA structure of *LlERF110*. (**B**) Sequence alignment of LlERF110 with ERF110 from *Arabidopsis thaliana*, *Arachis hypogaea*, *Daucus carota* var. *sativa*, and *Helianthus annuus*. AP2/ERF domain is indicated by red box, blue color means the conserved amino acid and red color means the difference among homologous protein from different species. (**C**) Phylogenetic tree of LlERF110 and ERF110s in other species. The ERF110 proteins were from: *Arachis duranensis* AdERF110 (XP_015942600.1), *Arachis hypogaea* AhERF110-like (XP_025621140.1), *A. thaliana* AtERF110 (At5g50080.1), *Cucumis melo* CmERF110 (NW_007546340.1), *Dendrobium catenatum* DcERF110-like (XP_020695086.2), *Daucus carota* DsERF110-like (XP_017229979.1), *Erythranthe guttata* EgERF110-like (XP_012829472.1), *Gossypium hirsutum* GhERF110-like (XP_016693790.1), *Gossypium raimondii* GrERF110-like (XP_012482409.1), *Helianthus annuus* HaERF110-like (XP_022019027.1), *Musa acuminata* subsp. *Malaccensis* MmERF110-like (XP_018682388.1), *Nicotiana attenuate* NaERF110-like (XP_019258503.1), *Nicotiana sylvestris* NsERF110-like (XP_009777819.1), *Solanum pennellii* SpERF110-like (XP_015060005.1), and *Solanum tuberosum* StERF110-like (XP_006361609.1). These trees were analyzed using TBtools v1.09876 and drawn using iTOL.

**Figure 4 ijms-23-16135-f004:**
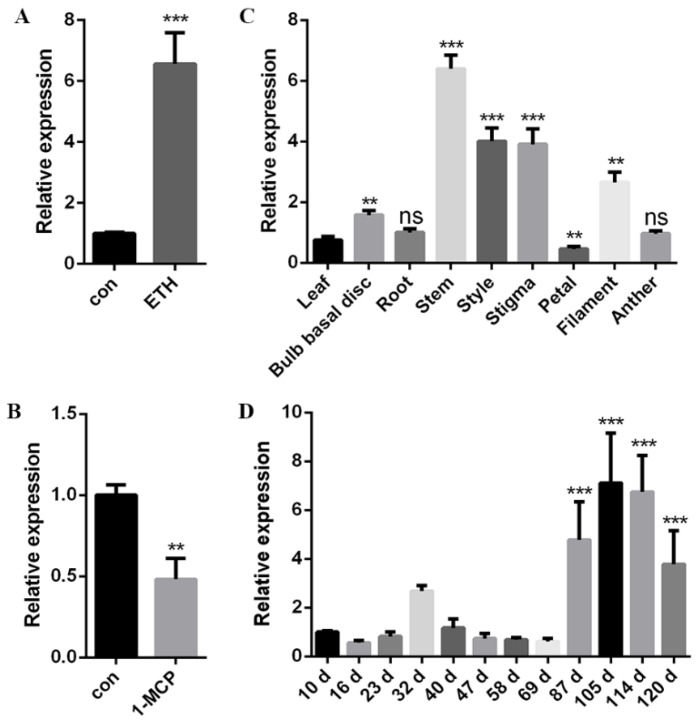
Expression analysis of *LlERF110* using qRT-PCR. (**A**) Relative expression of *LlERF110* with exogenous 2 ppm ETH treatment. (**B**) Relative expression of *LlERF110* with 2 ppm 1-MCP treatment. (**C**) Relative expression of *LlERF110* in different tissues of 120-day-old lily. (**D**) Relative expression of *LlERF110* at different growth stages, from 10 d to 120 d, in lily. 18S rRNA was used as an internal control. Three independent experiments were performed for each sample with three technical replicates, and the results of one experiment are shown in the figure. *T*-test analysis of variance was used to identify statistically between different treatment. Samples with different stars are significantly different: ** *p* < 0.01, *** *p* < 0.001, ns means non-significance.

**Figure 5 ijms-23-16135-f005:**
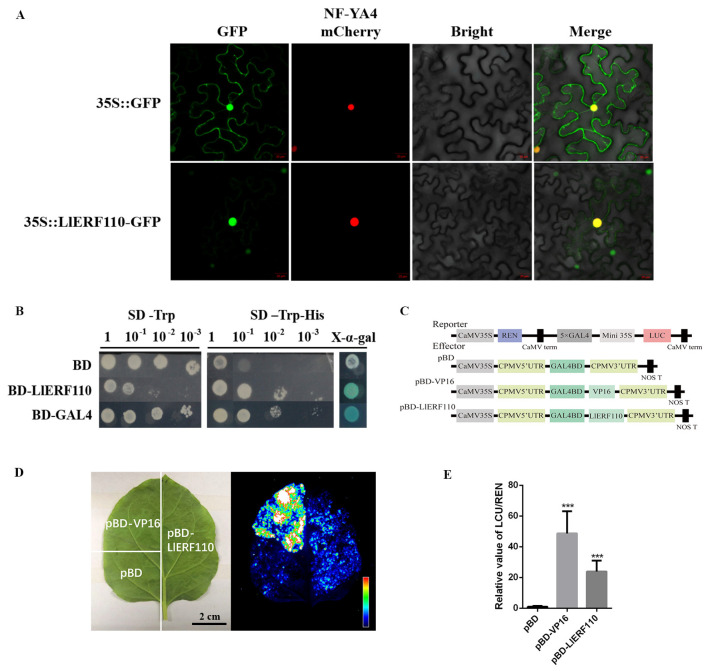
Subcellular localization and transactivation activity of LlERF110. (**A**) The subcellular localization of the LlERF110-eGFP fusion protein was examined in *N. benthamiana*. Bar: 20 µm. The plasmid 35S::GFP and P35S::LlERF110-GFP was co-infiltrated with the nuclear marker NF-YA4-mCherry into tobacco leaves, respectively. 35S::GFP was used as the negative control. Green and red fluorescence were visualized by confocal microscopy 72 h after infiltration. (**B**) Transactivation activity of LlERF110 was examined in yeast. The empty vector pGBKT7 (BD) was used as a negative control. Additionally, the pGBKT7 vector with a sequence encoded the yeast transcription activating protein GAL4 (BD-GAL4) was used as a positive control. The transformed yeast cells were grown in a synthetic dextrose minimal medium (SD) with or without histidine. The cultured yeast cells were diluted to different concentrations of 10^−1^, 10^−2^ and 10^−3^ to test their growth vitality. X-α-gal staining was used for the detection of β-gal activity of yeasts without dilution. (**C**) The pattern graph of reporter and effector constructs. The effector vector was named as pBD and used as the negative control, The coding sequence of LlERF110 with the termination codon was inserted into the pBD effector vector as pBD-LlERF110. pBD-VP16 was used as the positive control. (**D**) Live imaging of transcriptional activation activity of LlERF110 in *N. benthamiana* leaves. pBD-VP16 was used as a positive control. Bar: 2 cm. (**E**) Quantitative analysis of transcriptional activation activity of the LlERF110 protein in (**D**). *T*-test analysis of variance was employed to identify treatment means that differed statistically. Samples with different stars are significantly different; *** *p* < 0.001.

**Figure 6 ijms-23-16135-f006:**
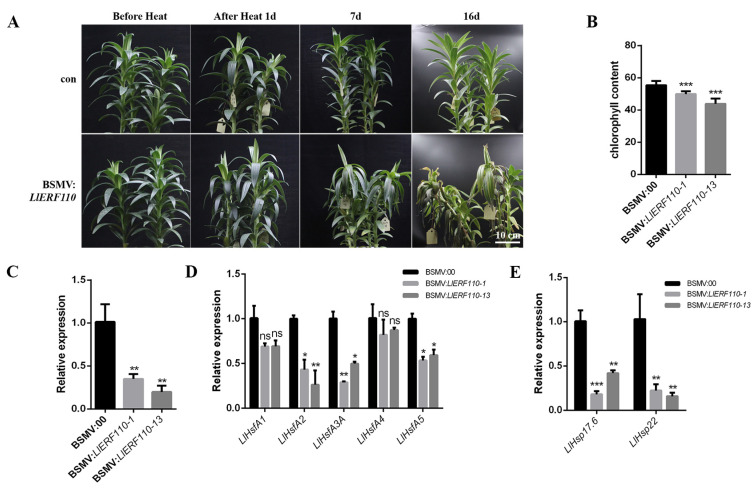
Phenotype of *LlERF110*-silenced plants under heat stress and expression of *HSFs* and *HSPs* in silenced plants. (**A**) The phenotypes of the *LlERF110*-silenced lines recovered at room temperature at 1 d, 7 d, and 16 d after treatment with heat stress at 42 °C for 24 h. Bar: 10 cm. (**B**) Chlorophyll content in leaves of *LlERF110*-silenced plants (BSMV: *LlERF110-1* and BSMV: *LlERF110-13*) recovered at room temperature for 8 d after heat stress. (**C**) Expression of *LlERF110* in silenced lines (BSMV: *LlERF110-1* and BSMV: *LlERF110-13*) using qRT-PCR. (**D**) Expression of *Hsfs* in *LlERF110* silenced lines. (**E**) Expression of *Hsps* in *LlERF110*-silenced lines (BSMV: *LlERF110-1* and BSMV: *LlERF110-13*). 18S rRNA was used as a control. Three independent experiments were performed, each with three technical replicates, the results of one experiment are shown. T-test analysis of variance was employed to identify treatment means that differed statistically. Samples with different stars are significantly different; * *p* < 0.05, ** *p* < 0.01, and *** *p* < 0.001; ns means non-significance.

**Figure 7 ijms-23-16135-f007:**
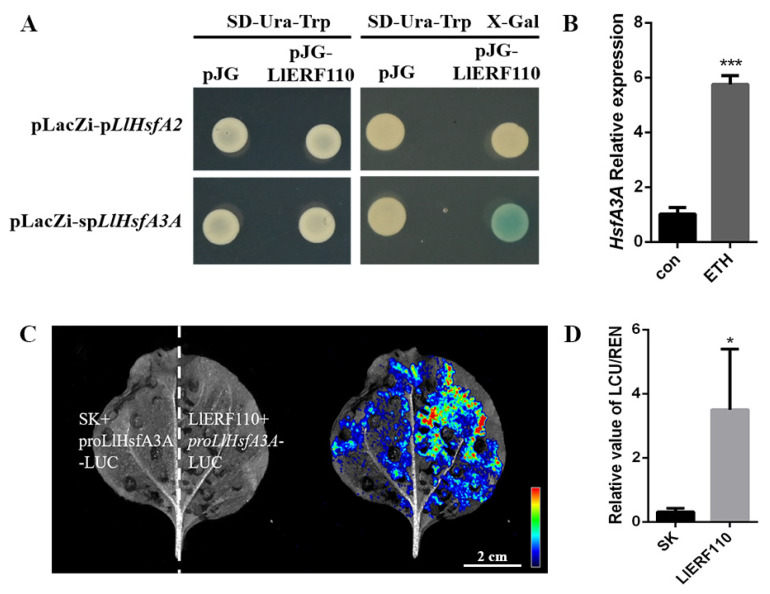
LlERF110 binds to the promoter of *LlHsfA3A* and activates its transactivation activity. (**A**) The bait (pLacZi-*pLlHsfA2* or *spLlHsfA3A*) and prey (pJG or pJG-LlERF110) constructs were cotransformed into the yeast strain Y187. The baits transformed with the prey pJG were used as negative control. Interaction between bait and prey was determined by cell growth on SD/-Ura-Trp containing X-Gal. (**B**) Relative expression of *LlHsfA3A* in lily leaves after exogenous 2 ppm ETH treatment measured using qRT-PCR. (**C**) The dual-luciferase reporter assay was used to detect the interaction between LlERF110 and the *LlHsfA3A* promoter in *N. benthamiana* leaves. A 1020-bp fragment of LlHsfA3A promoter was used in this assay. Empty vector pGreenII 62-SK (SK) and *ProLlHsfA3A*:LUC was used as negative control. Suspensions of *A. tumefaciens* contained different vectors that were mixed and infiltrated into *N. benthamiana* leaves, and LUC signals were detected 72 h after infiltration. Bar: 2 cm. (**D**) The ratio of LUC/REN of the empty vector (SK) co-transformed with the *proLlHsfA3A*-LUC vector was used as the calibrator (set as 1). Three independent experiments were performed. *T*-test analysis of variance was employed to identify treatment means that differed statistically. Samples with different stars are significantly different; * *p* < 0.05, and *** *p* < 0.001.

**Figure 8 ijms-23-16135-f008:**
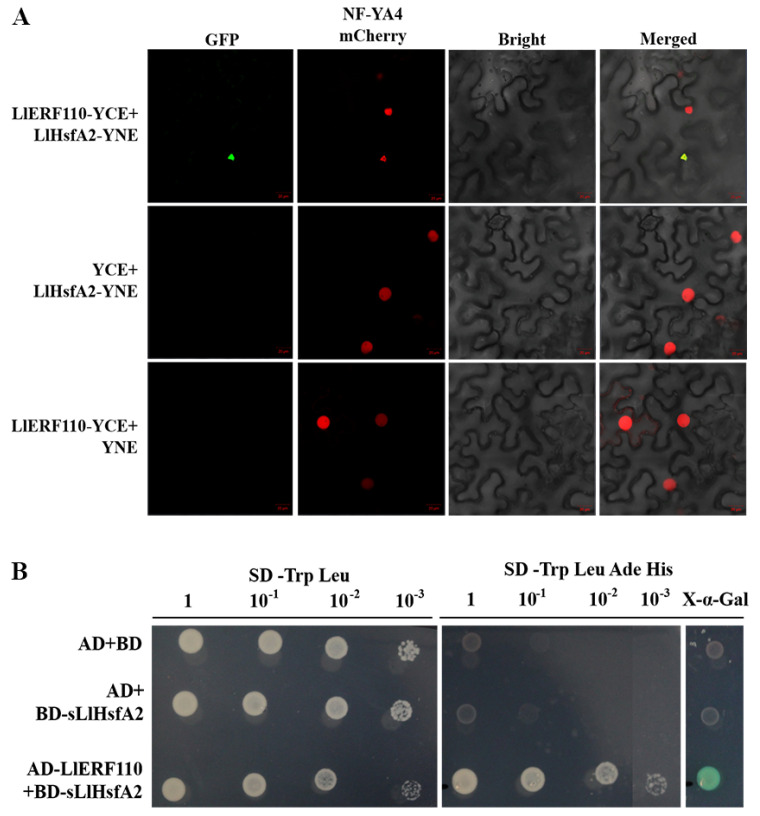
LlERF110 interacts with LlHsfA2. (**A**) BiFC assay. The plasmid LlERF110-pSPYCE (YCE) and HsfA2-pSPYNE (YNE) were co-infiltrated into the *N. benthamiana* leaves with the nuclear marker NF-YA4-mCherry, respectively. LlERF110-YCE together with YNE, HsfA2-YNE together with empty vector YCE were used as negative controls. Green and red fluorescence were visualized by confocal microscopy 72 h after infiltration. Bars = 20 μm. (**B**) Y2H assay. The empty vector pGADT7(AD) + pGBKT7 (BD) and pGADT7(AD) + BD-sLlHsfA2 was used as two negative controls. The transformed yeast cells were grown in an SD medium with or without adenine and histidine. Cultured yeast cells were diluted to different concentrations of 10^−1^, 10^−2^ and 10^−3^ to test their growth vitality. X-α-gal staining was used for the detection of β-gal activity of yeast without dilution.

**Figure 9 ijms-23-16135-f009:**
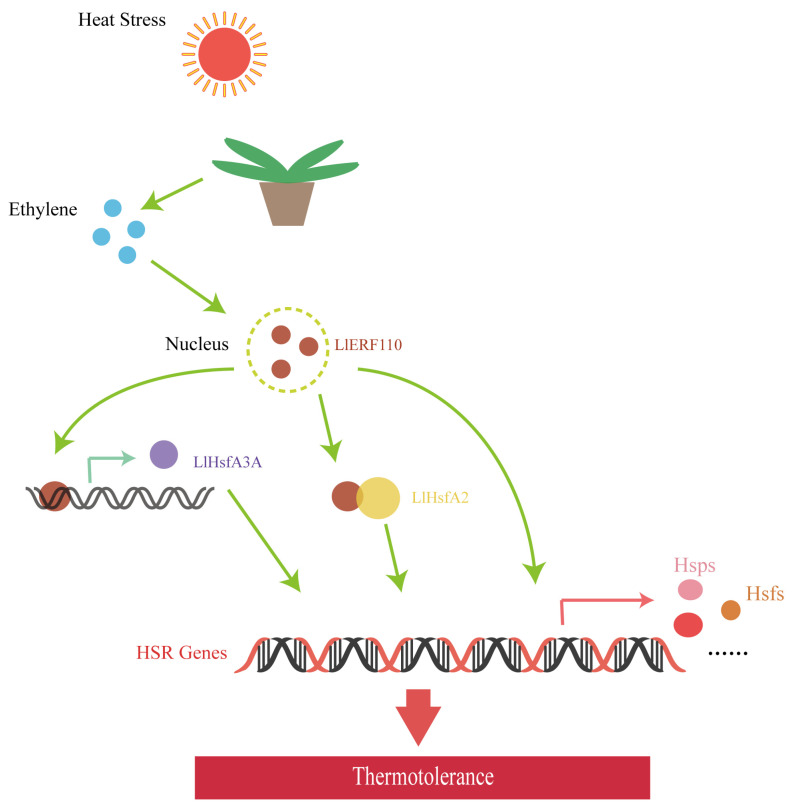
A working model of LlERF110 induces the expression of *LlHsfA3A*, and interacts with LlHsfA2 to regulate heat stress response in lilies.

## Data Availability

The authors confirm that the data supporting the findings of this study are available within the article & its Appendix A.

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
