# Peer review of "Ethylene Response Factor LlERF110 Mediates Heat Stress Response via Regulation of LlHsfA3A Expression and Interaction with LlHsfA2 in Lilies (Lilium longiflorum)"

_ijms, 2022, doi:10.3390/ijms232416135_

Round 1

Reviewer 1 Report

In this study, it was shown that exogenous ethylene treatment would enhance the heat resistance of lilies. During this process, the role of LlERF110, cloned and characterized from lily (Lilium longiflorum)  plant was revealed.  It was demonstrated that LlERF110 might be the junction of heat stress response pathway and ethylene signaling pathway by regulating Hsf-Hsp 32 pathway.

The objective of this paper was clearly presented in the introduction part. Appropriate experiments were chosen and the results were presented well.

Author Response

Thanks for your recognition of our article,we have revised the manuscript under your advice.

Reviewer 2 Report

The authors have a detailed study of molecular events around ethylene and heat stress in lilies.

Interesting that the justification is for lily flowers but the work is on leaves mainly  please address 

the presentation has extensive problems with english expression-  serious editing is needed to produce a quality product.

however the approaches are sound-  I request that each expression assay is accompanied by  a diagram showing the approach    so much background is assumed for the reader    In this version of the paper there are too few details to understand methods easily. Also controls are not outlined and discussed.  This is especially true for the brevity in the supplemental sections.  

Round 2

Reviewer 2 Report

thank you for your thorough revision  -- reads very well

the studies tell a strong story  through all the avenues that you followed   

Author Response

Dear Reviewer 2:

Thank you again for giving us very detailed and rigorous guidance for our manuscript! We are really impressed by your revisions and comments We have revised the problems you pointed out in the text one by one. The respond to your comment is as following, and the revised parts in the text have been marked in red.

(Results 2.8) or another factor is needed that is missing in this expression system?

Response: Thanks for your suggestions. HsfA2 is the important gene in the HSF-HSP pathway which is induced by the heat stress and regulates many targets such as HSPs, ROS relates genes. ERF110 binds to the HsfA2 promoter in Arabidopsis, However , we did no found the binding promoter of LlHsfA2 by LlERF110 and in lilies. So we examined the interaction between LlHsfA2 and LlERF110 to see regulation at the protein level. We will further study the interaction of LlERF110 and LlHsfA5 which is involved in ROS pathway in heat stress response.

The language was reworded according to your advice in manuscript.

Thanks again for your kind suggestion.